# Wear Analysis of Tibial Inserts Made of Highly Cross-Linked Polyethylene Supplemented with Dodecyl Gallate before and after Accelerated Aging

**DOI:** 10.3390/polym14235281

**Published:** 2022-12-03

**Authors:** Jian Su, Jianjun Wang, Shitong Yan, Min Zhang, Ningze Zhang, Yichao Luan, Cheng-Kung Cheng

**Affiliations:** 1School of Biological Science and Medical Engineering, Beihang University, Beijing 100083, China; 2Beijing Institute of Medical Device Testing, Beijing 101111, China; 3School of Biomedical Engineering, Shanghai Jiao Tong University, Shanghai 200240, China

**Keywords:** dodecyl gallate, highly cross-linked polyethylene, knee prosthesis, wear performance, material aging

## Abstract

The wear of the tibial insert is one of the primary factors leading to the failure of total knee arthroplasty. As materials age, their wear performance often degrades. Supplementing highly cross-linked polyethylene (HXLPE) with dodecyl gallate (DG) can improve the oxidation stability of tibial inserts for use in total knee arthroplasty (TKA). This study aimed to evaluate the wear resistance of HXLPE supplemented with DG (HXLPE-DG) tibial inserts before and after accelerated aging. HXLPE-DG tibial inserts were subjected to wear testing of up to 5 million loading cycles according to ISO 14243, and the resulting wear particles were analyzed according to ISO 17853. The wear rate, number, size, and shape of the wear particles were analyzed. The average wear rate of the unaged samples was 4.39 ± 0.75 mg/million cycles and was 3.22 ± 1.49 mg/million cycles for the aged samples. The unaged tibial inserts generated about 2.80 × 10^7^ particles/mL following the wear test, but this was considerably lower for the aged samples at about 1.35 × 10^7^ particles/mL. The average equivalent circle diameter (ECD) of the wear particles from the unaged samples was 0.13 μm (max: 0.80 μm; min: 0.04 μm), and it was 0.14 μm (max: 0.66 μm; min: 0.06 μm) from the aged samples. Moreover, 22.1% of the wear particles from the unaged samples had an aspect ratio (AR) of >4 (slender shape), while this was 15.4% for the aged samples. HXLPE-DG improves the wear performance of the material over time. HXLPE-DG is a novel material that has been demonstrated to have antiaging properties and high wear resistance, making it a promising candidate for use in TKA. Nevertheless, the results are preliminary and will be clarified in further studies.

## 1. Introduction

Total knee arthroplasty (TKA) is considered an effective surgical treatment for knee osteoarthritis [1,2]. Highly cross-linked polyethylene (HXLPE) is increasingly used in total hip arthroplasty (THA) and is gradually replacing conventional ultrahigh-molecular-weight polyethylene (UHMWPE) as the material of choice for bearings due to its excellent wear resistance. Further researchers concluded that the benefit of HXLPE in total knee arthroplasty (TKA) is unpredictable and remains controversial due to delamination and cracking [3]. It is well understood that the addition of reinforcing fibers greatly improves the stiffness and strength of polymeric matrix composites [4,5,6]. Researchers suggested that adding antioxidants can slow down the oxidation. Thus, to improve the oxidation stability of HXLPE, vitamin E is often added to the blend as an antioxidant [7]. However, since the molecular structure of vitamin E contains only one hydroxyl group, the loss of vitamin E during irradiation cross-linking can reduce the oxidation resistance of the material [8]. As an alternative, Fu J et al. proposed supplementing HXLPE with dodecyl gallate (HXLPE-DG) [8]. Such polyphenols have multiple phenolic hydroxyl groups and can offer better oxidation resistance than HXLPE supplemented with vitamin E (HXLPE-VE). HXLPE-DG has also been reported to have comparable mechanical properties and biocompatibility to HXLPE-VE [8].

In addition to the antioxidants used, aging is another important factor affecting the long-term use and service life of polyethylene. At present, wear resistance of polymers has been intensively studied, but the effect of aging on the tribological properties of polymers is still poorly studied [9]. As a novel highly cross-linked polyethylene material, few studies have investigated the wear properties of HXLPE-DG, particularly its performance before and after aging.

The purpose of this study was to investigate the wear rate of HXLPE-DG before and after aging using in vitro wear tests. The quantity, size, and morphology of polyethylene wear particles in the lubricating medium were recorded to analyze the wear resistance and resistance to oxidation of HXLPE-DG. The hypothesis of this study was that the wear resistance of HXLPE-DG would reduce after aging.

## 2. Materials and Methods

### 2.1. Samples

Eight prosthetic knee tibial inserts (Beijing Naton Technology Group Co., Ltd., Beijing, China) were investigated in this study (Figure 1). All inserts were made of HXLPE-DG and produced using the same processing technology, with a size of 83.1 mm × 54.1 mm × 18.0 mm and crosslink density of 224 mol/dm^3^. The tensile properties of the materials used in this study are summarized in Table 1 [8]. Eight samples were randomly assigned to two groups, one that would undergo aging and one that would not be aged and would serve as a baseline. In the unaged group, three samples were randomly selected as the test group and the remaining sample was used as a control. In the aged group, all samples were aged according to ASTM-F2003 [10], with the samples being stored for 14 days at a constant temperature of 70 °C and a pure oxygen pressure of 5 atmospheres. Then, three samples were randomly selected as the test group sample and the remaining sample was used as the control.

For the wear test, each sample was assembled with the same type and specification of femoral condyle and tibial tray, all of which were made of CoCrMo.

### 2.2. Test Equipment and Test Parameters

#### 2.2.1. Wear Test

The wear test was carried out on an AMTI knee simulator (ADL-K6-01, Advanced Mechanical Technology Inc., Watertown, MA, USA). The test process included steps for sample presoaking, maintaining constant weight, fixing, abrasion, cleaning, and weighing.

The wear test was run for five million cycles in accordance with ISO 14243-1 [11]. The specific test parameters are shown in Table 2. The test samples were assembled as Figure 2. The control samples did not undergo the wear test but were subjected to the same axial load as the test group. This was then used to correct the mass loss of the inserts when calculating the wear rate. Calf serum with a protein concentration of 20 g/L (added with 0.02% sodium azide) was used as the lubricating medium.

The gravimetric wear and wear rate were calculated according to ISO 14243-2 [12]. The gravimetric wear ($W_{n}$) referred to the net loss of mass from each test specimen after n loading cycles and was calculated using Equations (1) and (2). The gravimetric wear of all the test specimens was calculated after the following wear cycles during the wear simulation: 500,000, 1 million, 2 million, 3 million, 4 million, and 5 million cycles.
(1)$$W_{n}=m_{0}-m_{n}+S_{n}$$
(2)$$S_{n}=\bar{m_{n}}-\bar{m_{0}}$$

$m_{0}$—the mass of the test specimen before the wear test.

$m_{n}$—the mass of the test specimen after n loading cycles.

$S_{n}$—the increase in mass of the control specimen over the same period.

$\bar{m_{0}}$—the mass of the control specimen before the wear test.

$\bar{m_{n}}$—the mass of the control specimen after n loading cycles.

The mass loss of the inserts was calculated and linearly fitted. The linear relationship between the mass loss and number of cycles n was determined by Equation (3), and the wear rate a_G_ was calculated.
(3)$$W_{n}=a_{G}\times \;n+b$$

b—constant.

The wear rate was taken as the average of the three test samples.

#### 2.2.2. Wear Particles Analysis

The wear particles analysis test was carried out according to ISO 17853 [13].

First, 10 mL of the calf serum was continuously mixed with 40 mL of hydrochloric acid (37% volume fraction) for approximately 1 h at 50 °C. The fluid turned a slightly purple color. Then, 0.5 mL of the digestion solution was mixed with 100 mL of methanol.

The filtrate was filtered using a polycarbonate membrane with a pore size of 0.1 μm, and the filtered membrane was dried in a precision drying oven for 24 h.

Images were taken using a scanning electron microscope (SEM) (S-4800, Hitachi, Tokyo, Japan) at a magnification of 10,000× *g* and acceleration voltage of 10 kV.

The polyethylene wear particles were assessed in terms of quantity, size, and morphology using the image analysis software ImageJ (contributors worldwide, USA).

The total number of polyethylene wear particles was calculated using Equation (4).
(4)$$N=\bar{n}\times \frac{S}{A}$$

N is the quantity of polyethylene wear particles on the filter membrane; $\bar{n}$ is the quantity of polyethylene wear particles from SEM images; S is the area of the filter membrane; A is the area of each SEM image.

Knowing that the wear particles on each filter membrane were from 0.1 mL of the calf serum medium, the content of wear particles in the full volume of calf serum medium could then be calculated.

The size of the polyethylene wear particles was represented by the equivalent circle diameter (ECD) [14]. Since the wear particles had an irregular shape, the diameter of a sphere equal to the projected area of the wear particles was used to represent the particle size, calculated by Equation (5).
(5)$$\mathrm{ECD}={\left(4\times \frac{A}{\pi }\right)}^{\frac{1}{2}}$$

The shape of the polyethylene wear particles was characterized by the aspect ratio (AR) [14], which is the ratio of the maximum diameter to the minimum diameter of the projected surface of the wear particles, as shown in Equation (6).
(6)$$AR=d_{max}/d_{min}$$

## 3. Results

### 3.1. Wear Rate

The average wear rate of the unaged HXLPE-DG inserts was 4.39 mg/million cycles but was considerably lower for the aged HXLPE-DG inserts at 3.22 mg/million cycles (Table 3). The wear rate of the HXLPE-DG polyethylene inserts decreased by 26.7% after aging.

### 3.2. Wear Particles Analysis

SEM images of wear particles from the unaged and aged polyethylene inserts are shown in Figure 3. Immediately it can be seen that there are very few wear particles with an ECD greater than 1 μm, and the majority of particles are round or spherical in shape.

#### 3.2.1. Amount of Wear Particles

The average number of polyethylene wear particles on each filter membrane was calculated according to Equation (4), giving about 2.80 × 10^7^ particles/mL for the unaged samples and 1.35 × 10^7^ particles/mL for the aged samples (Figure 4). The serum from the wear test of the unaged polyethylene contained 107.4% more wear particles than the serum from the aged polyethylene.

#### 3.2.2. Size of Wear Particles

The average ECD of the wear particles from the unaged polyethylene was 0.13 μm (max: 0.80 μm; min: 0.04 μm), and the average ECD from the aged polyethylene was 0.14 μm (max: 0.66 μm; min: 0.06 μm) (Figure 5).

Wear particles from the unaged polyethylene within the size range of 0.00–0.20 μm had a distribution of 2.24 × 10^7^/mL, particles within the range of 0.21–0.40 μm were distributed at 5.12 × 10^6^/mL, within the range of 0.41–0.60 μm were 5.39 × 10^5^/mL, and no wear particles with an ECD of 0.60 μm or larger were found. Wear particles with an ECD of less than 0.4 μm accounted for 98.3% of all particles from the unaged polyethylene.

Similarly, wear particles from the aged polyethylene within the size range of 0.00–0.20 μm were distributed at 1.03 × 10^7^/mL, within the range of 0.21–0.40 μm were distributed at 2.20 × 10^6^/mL, within the range of 0.41–0.60 μm were distributed at 6.12 × 10^5^/mL, within 0.61–0.80 μm were 3.67 × 10^5^/mL, and no wear particles with an ECD greater than 0.80 μm were found (Figure 6). For the aged polyethylene inserts, 92.6% of the wear particles had an ECD of less than 0.4 μm. The aged polyethylene produced a greater quantity of larger particle sizes than the unaged polyethylene.

#### 3.2.3. Shape of Wear Particles

The shape of the HXLPE-DG wear particles was evaluated using the aspect ratio (AR). The average value of AR for the unaged wear particles was 2.8 (max: 9.9; min: 1.0), and the average value of AR for the aged wear particles was 2.7 (max: 7.7; min: 1.1) (Figure 7). The percentage of particles with AR > 4 (slender shape) decreased with aging, from 22.1% unaged to 15.4% aged.

## 4. Discussion

A previous study from our research group reported that unaged HXLPE-DG knee inserts had a wear rate of 3.92 mg/million cycles [1], which was similar to the results of this study, although the earlier study did not consider the aged specimen. Haider et al. reported UHMWPE tibial inserts with a wear rate of 19.88 mg/million cycles [15], which was more than four times that of HXLPE-DG. The surface of UHMWPE is primarily composed of unbranched linear structures (free radicals). When subjected to high temperature or long-term friction, the bonding force between the molecular chains of the polyethylene structure weaken, resulting in surface wear [16,17]. The polyphenol groups in HXLPE-DG can eliminate free radicals and reduce oxidation, thereby improving the wear resistance [8]. B.R. Micheli et al. reported HXLPE-VE tibial inserts with a wear rate of 2.4 mg/million cycles [18], and T.M. Grupp et al. reported a wear rate of 5.3–5.6 mg/million cycles [19]. The addition of dodecyl gallate to the insert material provides comparable improvements in the wear resistance to vitamin E.

The results of this study showed that subjecting HXLPE-DG to accelerated aging for 14 days reduced the wear rate by 26.7%. In contrast, Affatato et al. found that aging HXLPE-VE did not have a significant effect on the wear behavior [20,21]. Other studies showed that the oxidative induction time of polyethylene blended with polyphenols after aging treatment was greater than that of pure polyethylene, demonstrating the benefits of polyphenols on oxidative stability [22,23].

The wear rate of the three HXLPE-DG inserts after aging ranged from 1.88 mg/million cycles to 4.82 mg/million cycles, with a standard deviation of 1.49. There was a greater variation in the wear rate between the aged samples than the unaged samples. This shows that aging has a certain influence on the internal structure of polyethylene [24]. Further research may be needed to improve the stability of HXLPE-DG, in addition to refining the processing technology.

In total joint arthroplasty, aseptic implant loosening is a common cause of implant failure, resulting mainly from inflammatory reactions caused by implant wear particles [25,26]. Key factors in these osteolytic processes are the material type and the size and shape of the wear particles [27]. Studies have shown that the greater the number of particles, the greater the resulting macrophage response, and the number of submicron particles is critical [28,29]. This current study found that the number of wear particles per unit volume from the unaged polyethylene was 102.9% more than the aged polyethylene, demonstrating a considerable reduction in the number of wear particles after aging. Zhang et al. [30] pointed out that irradiation can generate free radicals inside polyethylene, and the free radicals are easily oxidized which can lead to degradation of the mechanical properties of the bulk material. When HXLPE blended with polyphenols is aged, some free radicals are preferentially combined with polyphenol groups, which increases the degree of cross-linking and increases the wear resistance of the polyethylene. This was one reason why the wear rate of HXLPE-DG reduced after aging [8].

ECD and AR are commonly used to measure particle size and morphology [14,31]. The difference in the average ECD of unaged polyethylene wear particles (0.13 μm) and aged wear particles (0.14 μm) was not significant. The wear particles produced from both groups were small, all less than 0.80 μm, and were granular in form, resembling the morphology of wear particles reported for high cross-linked polyethylene (HXLPE).

Markhoff et al. [31] reported that the ECD of most wear particles from UHMWPE, HXLPE, and HXLPE-VE was between 0.1 µm and 0.2 µm. In a previous study from our research group, the average ECD of wear particles from unaged HXLPE-DG was 0.18 μm after 3 million cycles [8], which was similar to the results of this study where 98.3% of the wear particles generated were less than 0.4 μm. The reason may be that the cross-linking process reduces the deformability of the polyethylene inserts, resulting in less fibrous particles [31,32], and larger polyethylene particles could be crushed under the action of cyclic stress. At the same time, the proportion of large strip-shaped, needle-shaped, and fibrous particles was greatly reduced [1].

In this study, although the AR of the polyethylene wear particles was similar before and after aging, the percentage of slender particles with AR > 4 decreased by 29.9% after aging. Previous studies have shown that slender particles can induce more severe inflammatory factor expression and macrophage infiltration [33,34]. This indicates that the wear particles generated by the abrasion of aged HXLPE-DG would not be easily phagocytosed by macrophages compared with unaged HXLPE-DG.

There are some limitations of this study that should be noted. Due to the long time required for in vitro wear testing and associated costs, the number of test samples in this study was small, and the size of the tibial inserts was larger. Analytical studies were also not carried out on samples with different specifications to obtain more data to verify the wear resistance of HXLPE-DG. There may be some variation between the results of this study and those of other studies quoted in this article due to differences in test equipment, conditions, methods, etc., which may lead to certain limitations in the comparison of test data.

## 5. Conclusions

Preconditioning the HXLPE-DG inserts through accelerated aging further improved their resistance to wearing. HXLPE-DG is a novel material that has been demonstrated to have antiaging properties and a high wear resistance, making it a promising candidate for use in TKA. Nevertheless, the results are preliminary and will be clarified in further studies.

## Figures and Tables

**Figure 1 polymers-14-05281-f001:**
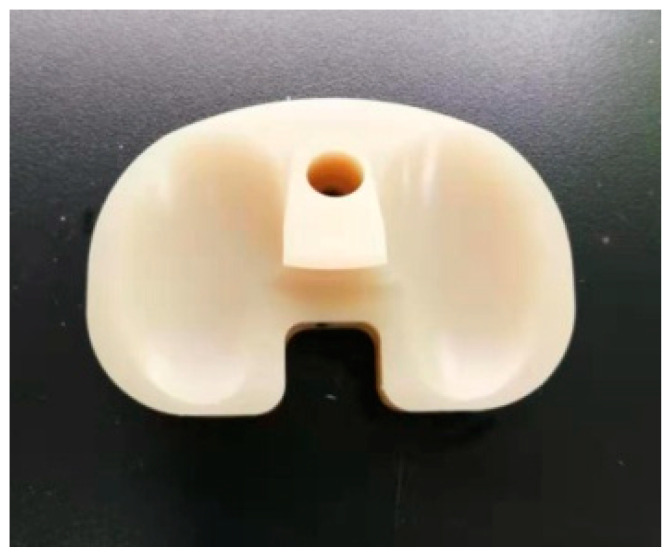
HXLPE-DG tibial insert.

**Figure 2 polymers-14-05281-f002:**
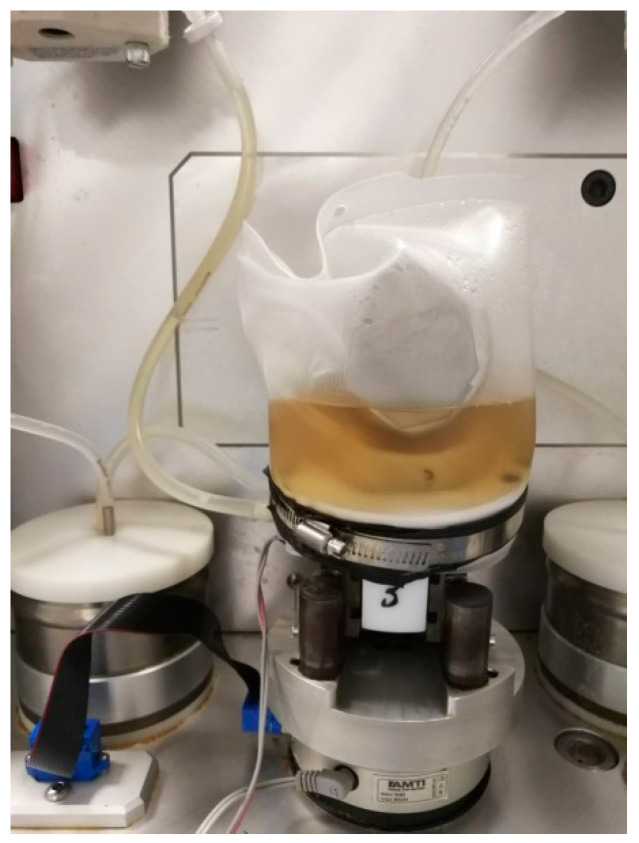
Setup of knee prosthesis for the wear test.

**Figure 3 polymers-14-05281-f003:**
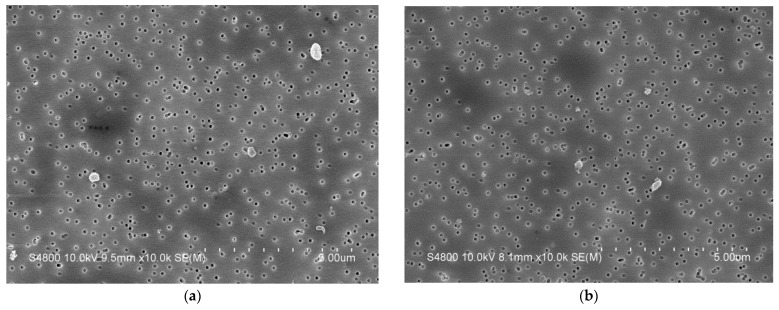
Image of polyethylene wear particles on filter membrane with pore size of 0.1 μm: (**a**) unaged; (**b**) aged.

**Figure 4 polymers-14-05281-f004:**
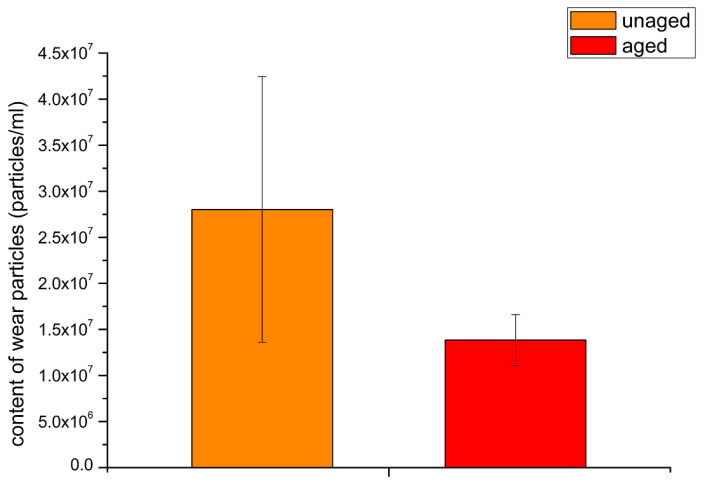
Amount of polyethylene wear particles.

**Figure 5 polymers-14-05281-f005:**
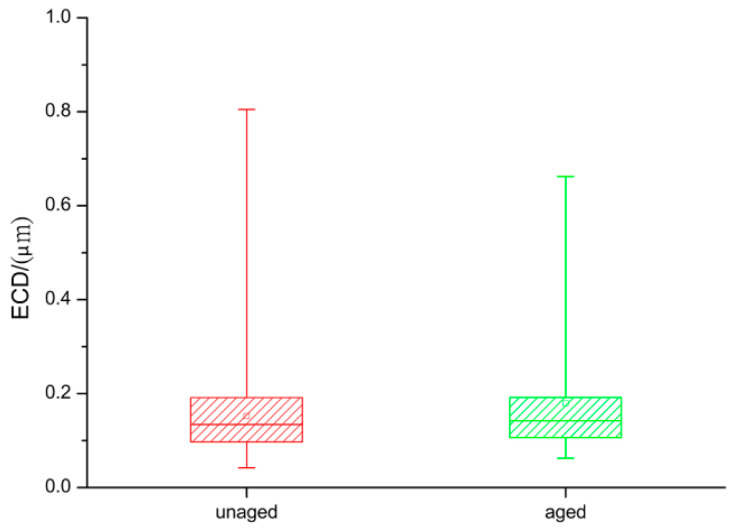
Size of polyethylene wear particles.

**Figure 6 polymers-14-05281-f006:**
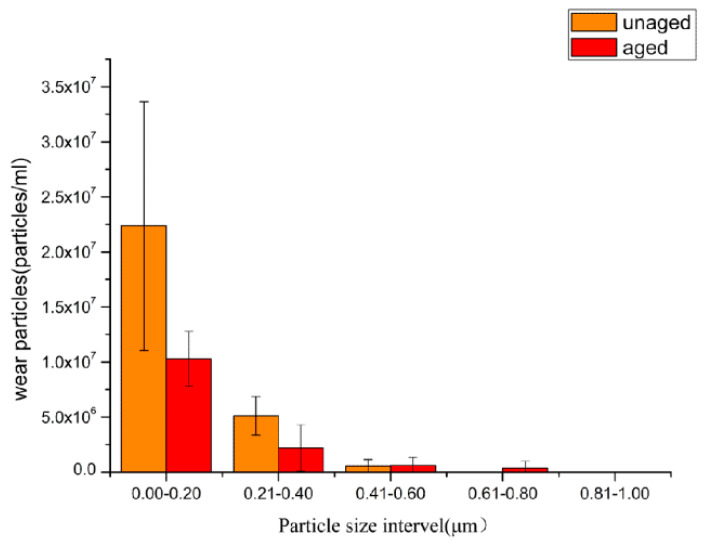
The size distribution of wear particles.

**Figure 7 polymers-14-05281-f007:**
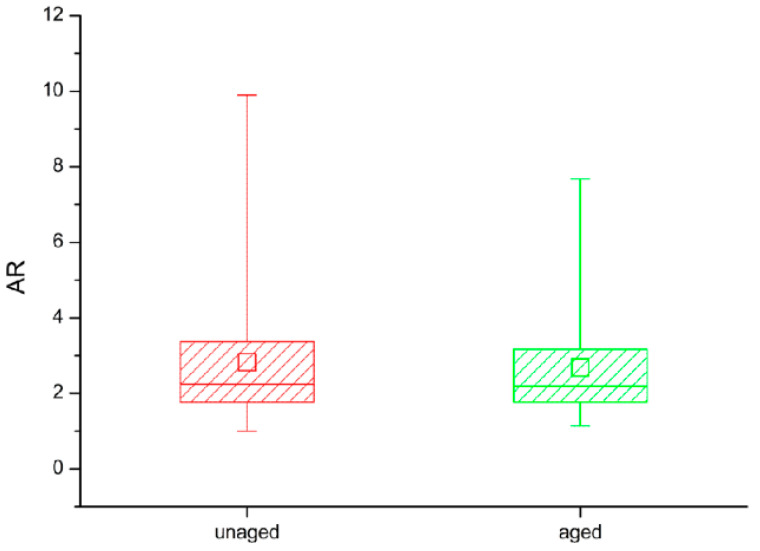
The shape of wear particles.

**Table 1 polymers-14-05281-t001:** The tensile properties of the materials.

	HXLPE-DG	
	Unaged	Aged
UTS (MPa)	60.6 ± 2.2	54.8 ± 5.4
EAB (%)	280 ± 14	261 ± 24
YS (MPa)	22.6 ± 0.3	23.8 ± 0.3
Impact strength (kJ·m^−2^)	77 ± 4	82 ± 1

**Table 2 polymers-14-05281-t002:** Wear test parameters.

Parameter	Test Specifications from ISO 14243-1
Load and displacement input curves	AP force: −265 N~110 N Flexion: 0°~58° Tibial rotation torque: −1 Nm~6 Nm Axial force: 168 N~2600 N
Test cycles	5,000,000
Test frequency	1 Hz
Test medium	Calf serum

**Table 3 polymers-14-05281-t003:** Wear rate after 5 million cycles.

Number	Preconditioning	Test Specimen Wear Rate (mg/Million Cycles)	Mean Wear Rate (mg/Million Cycles)	Standard Deviation
1	unaged	5.22	4.39	0.75
		3.78		
		4.17		
2	aged	1.88	3.22	1.49
		2.97		
		4.82		

## Data Availability

The data presented in this study is available on request from the corresponding author.

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
