# Peer review of "Wear Analysis of Tibial Inserts Made of Highly Cross-Linked Polyethylene Supplemented with Dodecyl Gallate before and after Accelerated Aging"

_polymers, 2022, doi:10.3390/polym14235281_

Round 1

Reviewer 1 Report (Previous Reviewer 2)

The authors have taken into account the previous comments and recommendations, the article can be published in present form

Author Response

Thank you. We have revised the manuscript accordingly.

Reviewer 2 Report (New Reviewer)

The paper presents an interesting approach based on the Wear Analysis of Tibial Inserts Made of Highly Cross-Linked Polyethylene Supplemented with Dodecyl Gallate Before and After Accelerated Aging. However, the innovation of the current research work should be further highlighted and emphasized. At the same time, the authors should consider the following comments to greatly improve the quality of the paper.

1. In the abstract, add a final statement that highlights the importance of this research and its possible potentials. Also, introduce the problem in the initial lines of the abstract.

2. The introduction needs to be improved by relating to the mechanics of the studied materials and their mechanical characteristics. The references to be included are: 10.1177/07316844211051733, 10.1016/j.polymertesting.2017.09.009, 10.1016/j.compstruct.2021.114698, 10.1177/0731684417727143, 10.1002/app.46770, 10.1016/j.porgcoat.2022.107015.

3. Kindly add a table that describes the main physical and chemical properties of the raw materials used in this study.

4. Were the preparation methods described by the authors come in accordance with a certain standard or do they follow previous procedures?

5. How can the average mass of each polyethylene particle be predicted in order to evaluate their number? 

6. How many samples were used per configuration for the wear test?

7. The conclusion needs to be modified to summarize the research outcomes in short statements with clear observations.

Round 2

Reviewer 2 Report (New Reviewer)

Most of the comments have been fulfilled. Kindly include these references: 10.1177/0731684417727143, 10.1002/app.46770, 10.1016/j.porgcoat.2022.107015.

This manuscript is a resubmission of an earlier submission. The following is a list of the peer review reports and author responses from that submission.

Round 1

Reviewer 1 Report

Dear authors,

Thanks for submitting your manuscript on wear behavior of highly cross-linked PE with dodecyl gallate. The results and discussion section needs to be further elaborated.

Abstract: HXLPE and HHXLPE are used for highly cross-lined PE. Authors should only use one abbreviation to describe the material in the manuscript.

Line 18/Abstract: Words describing the different sections (methods/results/conclusions) in the manuscript is not required in abstract.

Keywords: HXLPE is also described as text, therefore, abbreviation is not required.

Methods: mL and ml are used as units. mL should be only used throughout the manuscript.

Results & Discussions: The authors have only presented their numerical results and wrote what other authors have reported in previous studies in the discussions section. The authors have not critically addressed one of the consistently observed result: higher standard deviation in unaged samples vs. aged sample and the scientific reasons for this observation.

Lines 249-252: The authors have mentioned in the experimental section that ASTM/ISO test methods have been followed. One of the primary reason for applying international standards is to obtain comparable data. However, the authors have provided a disclaimer that their results may not be comparable. This is counter-intuitive.

Reviewer 2 Report

The work is devoted to the experimental study of the influence of aging of modified cross-linked polyethylene on its wear resistance as it is related to its use in prosthetics of knee joints, the relevance of the work does not raise any doubts. The results were obtained using normative and validated research methods.

I think that the work can be published with the following corrections and clarifications:

1.         As noted by the authors themselves, the study was performed on a small number of samples (Table 2 shows the results for three samples, not subjected to aging and three artificially aged samples). In my view, in this case it is necessary to focus on the fact that the results are preliminary, especially since the wear results in the group of artificially aged samples differs by several times

2.         The paper draws an unambiguous conclusion that the wear of unaged samples is about 27% higher than that of aged samples, which is made on the comparison of average wear values obtained on a small number of samples and with a large (especially for aged samples) standard deviation. In my view, this is not quite correct. For example, the maximum wear for one of the aged samples (4.82) is higher than the minimum for one of the unaged samples (3.78). In the abstract the conclusion about significantly reduced wear of aged specimens sounds as fully confirmed, I think that in the abstract it should be also mentioned that the results are preliminary and will be clarified in further studies.

3.         Samples not subjected to aging showed a much smaller variation in the results than samples after aging. It would be desirable to reflect in the article the authors' opinion on the possible reasons for this fact.

 4.         The abstract contains the following statement: "Conclusion: Supplementing HXLPE with DG improves the wear performance of the material over time." However, the article only examined DG-modified HXLPE samples and does not show the wear results of HXLPE without the addition of DG or, for example, with the addition of VE. On what basis is this statement made? In the paper, only UHMWPE was compared in terms of wear as an example.
